# Alteration of the Gut Microbiota in Pigs Infected with African Swine Fever Virus

**DOI:** 10.3390/vetsci10050360

**Published:** 2023-05-18

**Authors:** Young-Seung Ko, Dongseob Tark, Sung-Hyun Moon, Dae-Min Kim, Taek Geun Lee, Da-Yun Bae, Sun-Young Sunwoo, Yeonsu Oh, Ho-Seong Cho

**Affiliations:** 1Bio-Safety Research Institute, College of Veterinary Medicine, Jeonbuk National University, Iksan 54596, Republic of Korea; dudtmd3315@naver.com (Y.-S.K.); chunsu17@naver.com (S.-H.M.); shortm@naver.com (T.G.L.); bdy7700@gmail.com (D.-Y.B.); 2Korea Zoonosis Research Institute, Jeonbuk National University, Iksan 54531, Republic of Korea; tarkds@jbnu.ac.kr (D.T.); daeminkk@gmail.com (D.-M.K.); 3CARESIDE, Ltd., Seongnam 13209, Republic of Korea; sunwoosy@gmail.com; 4Institute of Veterinary Science, College of Veterinary Medicine, Kangwon National University, Chuncheon 24341, Republic of Korea

**Keywords:** African swine fever, ASF, gut microbiome, pig, next-generation sequencing

## Abstract

**Simple Summary:**

This study analyzed the dynamic changes in the intestinal microbiome of pigs after being infected with the highly virulent African swine fever virus (ASFV) genotype II strain. The fecal microbiomes of infected pigs were thoroughly investigated according to the four phases of ASFV infection: before infection, primary phase, clinical phase, and terminal phase. As a result, the richness indices (ACE and Chao1) significantly decreased in the terminal phase. The relative abundances of short-chain-fatty-acids-producing bacteria, such as *Ruminococcaceae*, *Roseburia*, and *Blautia*, significantly decreased during ASFV infection. On the other hand, the abundance of *Proteobacteria* and *Spirochaetes* increased. The predicted functional analysis using PICRUSt revealed a significantly reduced abundance of 15 immune-related pathways in the ASFV-infected pigs. This study provides evidence for further understanding the ASFV–pig interaction and suggests that changes in gut microbiome composition during ASFV infection may be associated with the status of immune modulation.

**Abstract:**

The factors that influence the pathogenicity of African swine fever (ASF) are still poorly understood, and the host’s immune response has been indicated as crucial. Although an increasing number of studies have shown that gut microbiota can control the progression of diseases caused by viral infections, it has not been characterized how the ASF virus (ASFV) changes a pig’s gut microbiome. This study analyzed the dynamic changes in the intestinal microbiome of pigs experimentally infected with the high-virulence ASFV genotype II strain (N = 4) or mock strain (N = 3). Daily fecal samples were collected from the pigs and distributed into the four phases (before infection, primary phase, clinical phase, and terminal phase) of ASF based on the individual clinical features of the pigs. The total DNA was extracted and the V4 region of the 16 s rRNA gene was amplified and sequenced on the Illumina platform. Richness indices (ACE and Chao1) were significantly decreased in the terminal phase of ASF infection. The relative abundances of short-chain-fatty-acids-producing bacteria, such as *Ruminococcaceae*, *Roseburia*, and *Blautia*, were decreased during ASFV infection. On the other hand, the abundance of *Proteobacteria* and *Spirochaetes* increased. Furthermore, predicted functional analysis using PICRUSt resulted in a significantly reduced abundance of 15 immune-related pathways in the ASFV-infected pigs. This study provides evidence for further understanding the ASFV–pig interaction and suggests that changes in gut microbiome composition during ASFV infection may be associated with the status of immunosuppression.

## 1. Introduction

African swine fever (ASF) is a devastating infectious disease in pigs and wild boars characterized by viral hemorrhagic fever [1]. The fatality rate of ASF has been reported to be as high as 100%, and it causes a great deal of economic damage globally [2], being listed as one of the notifiable diseases by the World Organization for Animal Health (WOAH). Furthermore, since there is no vaccine for ASF, animal slaughter remains almost the only method to control the disease [3].

ASFV first enters the pig’s body through the tonsils and/or dorsal pharyngeal mucosa. After passing through the mandibular or retropharyngeal lymph nodes, it spreads systemically through viremia. It can be detected in almost all pig tissues as it spreads [4]. The virus has a restricted cellular tropism and replicates primarily in macrophages and monocytes, which are important for viral persistence and dissemination [5,6]. Host cytokines produced by these infected cells play a crucial role in ASFV pathogenesis [7].

The clinical course of ASF can be influenced by various factors, such as the virus, the host, and the immunological status of the farm [4]. The course of the disease in domestic pigs can be categorized as peracute, acute, subacute, or chronic [4]. In Europe and Asia, the highly virulent strains responsible for ASF belong to genotype II, and they can cause acute to peracute disease with up to 100% lethality within 7–10 days. Clinical signs are typically non-specific and may include high fever, loss of appetite, gastrointestinal and respiratory symptoms, cyanosis, ataxia, and sudden death [8]. However, the underlying factors that influence the ASFV-related disease outcome and course are still not well understood. Several studies have pointed out that further research should focus on host responses against ASFV [8].

The gut microbiota is a complex and diverse microbial ecosystem that resides in the gastrointestinal (GI) tract of animals [9]. It contains numerous microorganisms such as bacteria, viruses, fungi, protozoa, and archaea, and plays a significant role in the development and maintenance of the immune system [10,11,12]. The absence of microbiota results in the incomplete development of the immune system, as seen in germ-free mice [13,14]. The gut microbiota can also regulate T-cell differentiation, mitigate excessive immune responses, and alleviate inflammation by producing potent metabolites, including short-chain fatty acids (SCFA) [15]. On the other hand, when gut epithelial integrity is lost, gut microbiota and their toxins, such as lipopolysaccharide (LPS) and incompletely digested fats and proteins, enter the bloodstream, leading to systemic inflammatory responses and tissue damage [16]. Thus, maintaining intestinal homeostasis and the microbial ecosystem is critical for the host’s overall health.

Increasingly, studies suggest that the disruption in the homeostasis of the GI microbiome and the host’s immune system can have negative effects on viral immunity [17]. Viruses can change the host’s gut microbiome without directly infecting the GI tract, which has been shown to control the severity of the disease [18,19]. Thus, microbiome change after viral infection is an important characteristic of the pathogen in the context of viral–host interactions. It has not been determined how ASFV changes the gut microbiome in a pig’s intestines. The objective of this study is to analyze the gut microbiome of pigs following infection with highly virulent ASFV. The study involved administering a genotype II strain of ASFV to pigs that were free of specific pathogens and observing changes in the gut microbiome. The goal was to gain insight into how ASFV affects the pig’s gut microbiota and identify the features of the modified microbiome. This will help determine if the altered microbiome contributes to the development and outcome of ASF.

## 2. Materials and Methods

### 2.1. Virus

The stock of the genotype II ASFV strain, South-Korea/19S804/2019/wild_boar, was obtained from the National Institute of Environmental Research, Republic of Korea. The inoculum was prepared by first diluting the stock virus 10-fold in Dulbecco’s Modified Eagle Medium (DMEM) (Gibco; Invitrogen, MA, USA) containing 1% antibiotic/antimycotic (10,000 units/mL of penicillin, 10,000 μg/mL of streptomycin, and 25 μg/mL of Fungizone^®^) (Gibco; Invitrogen, MA, USA). Subsequently, additional dilutions were performed to obtain 10^2^ 50% hemadsorbing doses per 2 mL (HAD 50/2 mL).

### 2.2. Animals

For the experiment, a total of 7 colostrum-fed, cross-bred (Yorkshire X Landrace X Durok) conventional piglets were purchased at 7 weeks of age from a commercial farm. At the time of arrival, they weighed between 18 and 22 kg, and both males and females were randomly mixed. These pigs were demonstrated to be free of porcine reproductive and respiratory syndrome virus (PRRSV) and porcine circovirus type 2 (PCV2) by RT-PCR and had no history of vaccination. They were housed and kept within the ventilated, totally slatted isolation cages for 7 days prior to the start of the experiments for acclimation. The animal procedures followed ethical guidelines and were approved by the Jeonbuk National University (JBNU) Institutional Biosafety Committee (IBC, Protocol #JBNU2022-03-001) and the Institutional Animal Care and Use Committee (IACUC, Protocol #JBNU 2022-028). All pigs had daily access to food and water throughout the experiments.

### 2.3. Animal Experiment

Pigs were divided into ASFV and NC groups (ASFV, challenge group, *n* = 4; NC, negative control group, *n* = 3) and moved to isolated rooms. Then, the ASFV group was inoculated with 2 mL of inoculum per animal within the right semimembranosus muscle, and the NC group was inoculated with 2 mL of phosphate-buffered saline (PBS). During the study, clinical evaluations, including rectal temperature measurements, were performed on all animals until their death. The evaluation followed the clinical scoring system as previously described for ASF [20]. The maximum score is 40 points. When the score was over 20, or the pigs indicated moribundity (moribund animals were defined as those that are expected to die within 24 h), euthanasia was performed [21]. They were inoculated with 6 mL of Alfaxalone (Alfaxan multidose, Careside, Seongnam, Republic of Korea) per animal to induce general anesthesia, followed by 3 mL of succinylcholine (Succipharm, Komipharm, Siheung, Republic of Korea), and their axillary and/or subclavian arteries were cut for bleeding.

### 2.4. Fecal DNA Extraction

Fresh fecal samples were collected every morning before feeding from both the NC and ASFV pig groups during the experiment. Fecal samples were taken into sterile cups shortly after pigs started defecating and stored at −80 °C immediately. The samples of ASFV-infected pigs were distributed into 4 phases of ASF (before infection, primary phase, clinical phase, and terminal phase) based on their clinical characteristics, according to the criteria previously described [22]. Briefly, fecal samples from pigs that did not exhibit fever or clinical symptoms after ASFV infection (referred to as the incubation period) were categorized as the primary phase. Additionally, samples from pigs with fever (>40 °C) and clinical symptoms (>10% of the maximum score) were categorized as the clinical phase, while samples collected two days prior to death were categorized as the terminal phase. The sample distribution according to the clinical evaluation is listed in Table 1. The sample was aliquoted into 200 mg, added to stool lysis (SL) buffer and proteinase K, and taken out of ABL-3. Genomic DNA was extracted from the sample by using AccuPrep^®^ (Bioneer, Daejeon, Republic of Korea) following the manufacturer’s instructions. Briefly, the process of extraction is as follows: incubated at 60 °C for 10 min and then centrifuged at 13,000 rpm for 5 min; added to sample binding (SB) buffer and incubated at 60 °C for 10 min; combined with isopropanol and centrifuged at 8000 rpm for one minute; after washing 2 times, centrifuged at 8000 rpm and then additional centrifugation at 13,000 rpm for a minute to make sure the alcohol is completely removed; and, finally, the sample DNA was extracted by adding 50 μL of the elution buffer. DNA quality was evaluated using an Epoch microplate spectrophotometer (BioTek Instruments, Winooski, VT, USA) and then the samples were stored at −20 °C until Next-Generation Sequencing.

### 2.5. Next-Generation Sequencing

PCR was performed for each fecal sample at total reaction volumes of 25 μL containing 2.5 μL of 10X Ex Taq buffer, 2 μL of 2.5 mM dNTP mix, 0.25 μL of Takara Ex Taq DNA Polymerase (5 U/μL), 2 μL of primer pair 515F-806R (10 pM, respectively), and 4 μL of the genomic DNA (gDNA) of the sample. The PCR primers flanked the V4 hypervariable region of the bacterial 16S rRNAs, and their sequences are represented in Table A1. The targeted gene was amplified in a Veriti™ 96-Well Fast Thermal Cycler (Applied biosystems, Woburn, MA, USA). The PCR conditions are as follows: initial denaturation at 95 °C for 3 min, followed by 25 cycles of denaturation at 95 °C for 30 s, primer annealing at 55 °C for 30 s, and extension at 72 °C for 30 s, with a final elongation at 72 °C for 5 min. The PCR products were purified with magnetic bead-based Agencourt AMPure XP Reagent (Beckman Coulter Inc., Brea, CA, USA) and DNA quality was evaluated using 1% agarose gel electrophoresis. The final DNA concentration was determined using a Qubit 2.0 fluorometer (Thermo Fisher Scientific Inc., Waltham, MA, USA). Mixed amplicons were pooled in equimolar amounts. Single-end sequencing (1 × 300 bp) was carried out with an Illumina iSeq Sequencing system (Illumina, San Diego, CA, USA) according to the manufacturer’s instructions [23,24].

### 2.6. Taxonomic Assignment of Sequence Reads

The output data from the Illumina iSeq sequencing system were analyzed using the EzBioCloud 16S database (CJ Bioscience, Seoul, Republic of Korea) [25] and the 16S microbiome pipeline for data processing, statistical analysis, and data graphing.

Briefly, the single-end raw reads were uploaded to the EzBioCloud 16S rRNA gene-based Microbiome Taxonomic Profiling (MTP) app. For primary processing, quality checking (QC) was carried out, and the low-quality sequences (<80 bp or >2000 bp and <Q25) were filtered out. The denoising and extraction of the non-redundant reads were conducted using DUDE-Seq software. The UCHIME algorithm was applied to detect and remove chimera sequences. Taxonomic assignment was performed using the VSEARCH program [26], which searched and calculated the sequence similarities of the queried single-end reads against the EzBioCloud 16S database. Species-level identification was determined using a cut-off of 97% similarity. Other cut-off values for higher taxonomic ranks are listed in Table A2. Sequences that did not match at the 97% similarity level were further clustered using the UCLUST tool with a similarity boundary of 97%. Consequently, the single-end reads obtained from each sample were assigned to various operational taxonomic units (OTUs).

### 2.7. Statistical Analysis

The differences in the gut microbiota between the 2 groups (NC and ASFV) during the 4 phases of ASFV infection (before infection, primary phase, clinical phase, and terminal phase) were investigated. The distribution of shared OTUs was compared using the Venny 2.1.0 server. To identify significant differences in alpha diversity, including the richness and diversity indices, across the four phases of ASFV infection, we employed generalized linear model (GLM) analysis and Bonferroni and Tukey post hoc tests using IBM SPSS ver 26 (IBM Corp., New York, NY, USA). To confirm differences in beta diversity among groups, we conducted Principal Coordinate Analysis (PCoA) based on Generalized UniFrac. Statistical significance for observed variations was assessed using the Permutational Multivariate Analysis of Variance (PERMANOVA) function with 999 permutations.

To investigate the taxa with significantly different relative abundance in the four phases, we utilized the single-factor analysis of the Microbiomeanalyst R package with the DESeq2 method (FDR < 0.05). Additionally, differential abundances of gut microbial composition were analyzed using a Random Forest classification. For the analysis, features with at least 4 reads and a 10% minimum prevalence across samples were included, and the data were transformed using a centered log ratio (CLR). To predict functional abundances, we used PICRUSt and annotated them using the Kyoto Encyclopedia of Genes and Genomes (KEGG) pathway database. Statistical significance in differentially abundant functional pathways within the gut microbiome between the NC and ASFV groups was determined through a t-test (*p* < 0.05). All *p*-values were corrected for a false discovery rate (FDR) of 0.05, and an FDR-corrected *p*-value below 0.05 was considered significant.

## 3. Results

### 3.1. Microbiome Analysis

#### 3.1.1. Characteristics of Sequencing Data

We obtained an average of 51,463 high-quality reads (average read length of 281 bp) from the four phases, namely, before infection, primary phase, clinical phase, and terminal phase. These reads resulted in 983 ± 70, 850 ± 121, 939 ± 76, and 818 ± 93 bacterial OTUs, respectively.

A total of 735 OTUs were matched in all groups, and 57, 72, 95, and 79 OTUs were uniquely identified in the before infection, primary phase, clinical phase, and terminal phase, respectively, as shown in Figure 1. The richness estimates (ACE, Chao1) were highest before infection with the ASFV and significantly decreased over the following phases (Figure 2a). On the other hand, the diversity estimates, such as NP Shannon and Shannon, significantly increased in the clinical phase, while the Simpson index significantly decreased (Figure 2b; Table 2).

Principal coordinates analysis (PCoA) was used to calculate beta diversity based on generalized UniFrac distances. The resulting PCoA scatterplot showed a clear separation between the negative control (NC) and ASFV-infected (ASFV) groups of pigs (Figure 3a). Additionally, structural segregation among the four phases was observed: before infection and NC, primary, clinical, and terminal phases (Figure 3b). These patterns were further confirmed by PERMANOVA, which indicated significant differences in gut microbiota composition among the compared groups (*p* < 0.001). Clustering trees were built using UPGMA clustering based on generalized UniFrac distance matrices (Figure 4). The tree showed that the composition of the gut microbiota in the four phases of ASFV infection clustered separately in each pig.

#### 3.1.2. Composition Analysis (Community Bar Plot)

The gut microbiotas of ASFV-infected pigs collected at four different phases were characterized to evaluate the variability. Relative abundance (%) was used to identify differentially abundant phyla, families, and genera among the phases in the ASFV (Figure 5) and NC group (Figure A1). For each rank, bar graphs representing the mean relative abundance among different phases can be found in Figure A3, Figure A4 and Figure A5. Additionally, using the DESeq2 method at the phylum, family, and genus level, a total of 91 features that were significantly different across the four phases during ASFV infection were identified and listed in Table A3. *Firmicutes* was the predominant phylum found in all phases. The dynamic change was mainly associated with a significant, steady increase in *Proteobacteria* and a decrease in *Actinobacteria* during ASFV infection (FDR < 0.001). The relative abundance of *Tenericutes* significantly decreased, while the *Spirochaetes* increased in the clinical and terminal phases (FDR < 0.05) (Figure 5a and Figure 6a).

Furthermore, *Ruminococcaceae* was generally the predominant family found in all phases. Major changes in family level were associated with a significant increase in *Bacteroidaceae* and *Succinivibrionaceae* and a decrease in *Lactobacillaceae, PAC001057_f* (*Mollicutes*), and *Coriobacteriaceae* during ASFV infection (FDR < 0.05) (Figure 5b and Figure 6b).

At the genus level, although *Sodaliphilus* was the predominant taxon before infection, *Prevotella* was predominantly found in the primary and clinical phases, and *Eubacterium_g23* was predominant in the terminal phase. Throughout all phases, the relative abundances of *Spirochaetaceae_uc*, *AF371579_g* (*Lachnospiraceae*), *Bacteroides*, and *FMWZ_g* were significantly decreased, while those of *PAC000683_g* (*Ruminococcaceae*), *EU463156_g* (*Bacteroidales*), Libanicoccus, and *Lactobacillaceae_uc* tended to decrease significantly (*p* < 0.05) (Figure 5c and Figure A2).

#### 3.1.3. Random Forest Analysis

Random forest classification was used to determine the significantly different taxa among the four phases of ASFV infection (Figure 7). A total of 15 taxa were found to be significantly different among the phases, with a Mean Decrease Accuracy of > 0.0015. Prior to ASFV infection, one family (Mollicutes_PAC001057_f), two genera (Slackia, Eubacterium_g23), and two species (*Senegalimassilia anaerobia*, *Blautia Obeum*) were significantly higher. During the primary phase, one phylum (Firmicutes), one order (*Lactobacillales*), and one genus (*Lactobacillus*) were found to be enriched. In the clinical phase, one genus (Erysipelotrichaceae_uc) and one species (*Bacteroidales_EU462269_s*) were significantly higher than the others. Finally, during the terminal phase, one phylum (*Proteobacteria*), one genus (*Succinivibiro*), and three species (*Succinivibrio_FJ680264_s*, *Treponema succinifaciens*, *Lachnospiraceae_PAC001296_s*) were found to be significantly higher compared to the other groups (Figure 7).

#### 3.1.4. Differences of Predicted Immune System-Related Function

The differences in the predicted functional pathways of the gut microbiota between the NC and ASFV groups were analyzed by using PICRUSt based on the metagenome prediction. Among the 22 level 3 KEGG pathways associated with the immune system, a total of 16 functional pathways showed significant changes (Figure 8, Table 3). The ASFV group had a decreased proportion compared with the NC group in the 15 pathways, including hematopoietic cell lineage, Th17 cell differentiation, Th1 and Th2 cell differentiation, the Toll and Imd signaling pathways, the Toll-like receptor signaling pathway, the Fc epsilon RI signaling pathway, the T-cell receptor signaling pathway, the B-cell receptor signaling pathway, the RIG-I-like receptor signaling pathway, the NOD-like receptor signaling pathway, the IL-17 signaling pathway, the chemokine signaling pathway, the intestinal immune network for IgA production, platelet activation, and leukocyte transendothelial migration (FDR < 0.05). On the other hand, the only pathway of antigen processing and presentation was higher in the ASFV group than in the NC group (FDR < 0.01).

## 4. Discussion

ASF is a representative viral hemorrhagic fever (VHF) in animals. In humans, all causative agents of VHF are classified as RNA viruses and are categorized into four families: Arenaviridae, Bunyaviridae, Filoviridae, and Flaviviridae [27]. However, in animals, VHFs are caused by a much wider variety of viruses, some of which have double-stranded RNA genomes and even DNA genomes [28]. A common feature of VHFs is that the viruses infect and replicate in monocytes–macrophages, producing pro-inflammatory cytokines [1,27]. In addition, many of them present with GI signs rather than respiratory signs, along with high fever. Particularly, some VHFs, such as Ebola and Marburg fever, exhibit the bystander apoptosis of uninfected lymphocytes, which is an important feature of the pathology of ASF [1,29]. Since lymphocytes are the main inducers and effectors of GI immunity [30,31], the depletion of lymphoids is likely to alter the gut microbiome negatively. This may lead to a vicious cycle via the depression of gut microbiome function and increased intestinal permeability, strengthening the pathogenesis of the disease. Microbiome changes during VHF infection have scarcely been studied, and a fundamental discussion remains as to whether the host immune mechanism associated with the microbiome can affect the pathogenicity and severity of VHFs. This study investigated how ASFV changes a normal pig’s gut microbiome and whether the altered microbiome due to ASFV infection could function in a beneficial or harmful way in terms of the pathology of the disease.

ASFV usually infects mononuclear macrophages, and the development of the disease is induced by the cytokines they release. The key features of the pathogenesis of ASF in domestic swine are as follows: (a) severe lymphoid depletion, including lymphopenia and a state of immunodeficiency, and (b) vast hemorrhages [32]. The GI tract contains a significant amount of lymphoid tissue, which is required to maintain gut immunity and homeostasis. This tissue is the most affected body site during ASFV infection. Accordingly, the intestinal environment is likely modified, resulting in alterations to the microbial ecosystem. The gut microbiome can play a positive role in developing and maintaining host immunity, or their opportunistic pathogenic properties can cause systemic inflammation as a double-edged sword [33].

In this study, a remarkable change was observed in the normal pig’s gut microbiome during ASFV infection, wherein the host became potentially susceptible to inflammation and immunodeficiency. The clear separation between the NC and ASFV groups in terms of the PCoA adequately reflects the Anna Karenina principle, implying that dysbiotic individuals vary more in microbial community composition than healthy individuals—paralleling Leo Tolstoy’s dictum that “all happy families look alike; each unhappy family is unhappy in its own way” [34]. This indicates that ASFV can induce certain perturbations within a healthy gut microbiome that generally require a lot of maintenance and result in time-course-varied patterns in individuals.

Wang et al. [22] suggested that the course of acute ASFV infection could be divided into three phases: the primary phase (0–2 dpi) without changes in serum cytokine levels or clinical symptoms; the clinical phase (3–7 dpi) characterized by progressive clinical features, the upregulation of various pro-inflammatory cytokines (e.g., TNF-α, IFN-α, IL-1β, IL-6), and sustained fever; and the terminal phase, marked by an additional sharp increase in multiple cytokines (TNF-α, IL-1β, IL-6, IL-8, and IL-10) and the partial recovery of IFN-α. Our study was conducted according to these phases; the sequence samples were grouped into each phase according to the pig’s clinical characteristics and a few outliers were removed. The four phases were clearly distinguished on the PCoA, indicating that the gut microbiome may be associated with disease development. The detailed mechanisms remain to be further revealed. In addition, specific bacterial groups moved significantly during each phase: various SCFA-producing bacteria changed significantly. SCFA is mainly produced by some members of Firmicutes and Bacteroidetes, which metabolize indigestible polysaccharides. Acetate, propionate, and butyrate are the major SCFAs produced in the gut [35]. SCFAs directly affect T-cell differentiation into effector T cells, such as Th1 and Th17 cells, as well as IL-10+ regulatory T cells (Treg), and have anti-inflammatory properties mediated through the G-protein-coupled receptor (GPCR) [36,37]. Butyrate, the main source of energy for colonic epithelial cells, inhibits the mRNA expression of pro-inflammatory cytokines in the mucosa by inhibiting NF-κB activation [35,38]. Butyrate, as a histone deacetylase inhibitor, can also alter gene expression, inhibit cell proliferation, and induce cell differentiation or apoptosis, leading to butyrate’s anti-inflammatory and anti-tumor properties [39]. Therefore, a decrease in the microbiota that produces butyrate and other SCFAs is likely to be associated with host immune system abnormalities.

Overall, the major SCFA-producing bacteria *Firmicutes* decreased during ASFV infection. *Ruminococcaceae,* including a number of SCFA-producing bacteria, was the predominant family in all phases of ASFV infection. The relative abundance of *Ruminococcaceae* progressively decreased, along with nine genera significantly reduced (FDR < 0.05). *Eubacterium_g23* was most involved in this change. The genus *Eubacterium* is composed of phylogenetically and phenotypically diverse species, and many of them produce butyrate [40]. *Subdoligranulum,* other butyrate producers within the same family, significantly decreased during ASFV infection as well (FDR < 0.05) [41]. In the family *Lachnospiraceae,* which is the second-largest portion in Firmicutes, *Blautia* has been shown to significantly decrease during ASFV infection. *Blautia* plays an important role in maintaining balance in the intestinal environment, upregulating intestinal Treg cells and preventing inflammation, and its reduced abundance has been associated with inflammatory bowel disease (IBD) patients [15,42]. In addition, the *Collinsella aerofaciens* group, a unique butyrate-producing bacterium in the phylum *Actinobacteria,* was also observed to decrease significantly during ASF infection [43]. Overall, this decrease in butyrate-producing bacteria may be associated with the exacerbation of ASF.

Meanwhile, in the phylum *Bacteroidetes,* some SCFA-producing bacteria such as *Alloprevotella, Bacteroides,* and *Parabateroides* were observed to increase significantly during ASFV infection. *Prevotella* increased after ASFV infection and was the most abundant genus of *Bacteroidetes* from the primary to terminal phases. *Alloprevotella* is recognized as a beneficial bacteria and can produce SCFA-containing acetate and butyrate and promote an anti-inflammatory environment [44,45,46]. *Bacteroides* and *Parabacteroides* have similar physiological characteristics regarding carbohydrate metabolism and secreting SCFAs. They are considered to play a key role in regulating host immunity [47]. For example, *B. fragilis* expresses the capsular polysaccharide A (PSA) to induce CD4+ T-cell-dependent immune response and activates immunomodulatory IL-10, exhibiting anti-inflammatory effects during herpes simplex encephalitis [48,49]. *P. distanosis* can regulate innate inflammatory responses by locking the release of TNF-α, IL-6, IL-17, IL-12, or IFN-γ and protect intestinal permeability by promoting intestinal succinate and secondary bile acid production [49]. These increases in beneficial bacteria suggest that they may be major symbiotic bacteria regulating immunity in the clinical and terminal phases of ASF. However, several microbes in *Bateroides* and *Parabacteroides* and their toxins have been pointed out as opportunistic pathogenic characteristics [49,50], and there is also a possibility that they will further worsen the disease progression of ASF. For instance, *Bacteroides* spp. normally enters aseptic tissue through the intestinal mucosa, eventually causing other disease conditions and even forming abscesses in the central nervous system [51,52]. In addition, *Alloprevotella, Bacteroides*, and *Parabacteroides* are the main succinate producers in the host intestine [45]. Succinate is recognized as a microorganism-derived metabolite associated with dysbiosis-related diseases such as obesity and IBD [53]. As shown by the progressive increase in the *Phascalctobacterium succinatutens* group after ASFV infection in DESeq2 analysis (FDR < 0.05), which only uses succinate as an energy source phase, the aforementioned bacteria can modify the intestinal environment to a succinate-rich environment during ASF.

One of the important results of this study is the microbiome change in the primary phase. The richness of the bacterial community significantly decreased in the primary phase of ASFV infection. For the cause of this observation, though environmental effects cannot be totally excluded, it is necessary to examine the possibility of the virus’s effects on the richness of the microbiome. It took only about 2 to 3 days for ASFV to be detected in the bloodstream and a few days more to observe the expression of host clinical signs, including fever [54]. ASFV itself and/or immune cells affected by ASFV that reach the intestine via blood circulation may cause significant changes in the intestinal microbial ecosystem before host clinical symptoms appear. To the authors’ knowledge, this is the first evidence that a virus can change the gut microbiome during the incubation period of the disease.

The altered microbiome resulting from ASFV infection is similar to that observed with PRRSV and severe fever with thrombocytopenia syndrome virus (SFTSV) infections reported elsewhere. The microbiome affected by the viruses shared several features regarding the increased abundance of *Proteobacteria* and *Spirochaetes* but also decreased SCFA-producing families of *Ruminococcaceae* and *Lachnospiraceae* [18,19]. These may be the major changes in which a pig’s gut microbiome is affected by viruses that infect immune cells. On the other hand, pig intestines affected by enteric viruses, such as porcine epidemic diarrhea virus (PEDV), were observed to have an increased abundance of *Escherichia-Shigella*, *Enterococcus*, *Fusobacterium*, and *Veillonella* and decreased *Bacteroidetes* such as *Bacteroides* and *Prevotella* [55,56]. Therefore, the microbiome can be controlled according to the mechanism that the virus uses for its infection and proliferation. Furthermore, the PRRSV-infected pigs in previous studies have shown a different microbiome profile in a strain-virulence-dependent fashion [19]. Future studies need to investigate the effect of the virulence of ASFV on the gut microbiome or vice versa.

In the predictive functional analysis performed using PICRUSt, the immune-related pathways of the gut microbiome in the ASFV group were significantly compromised, indicating that ASFV modified the gut microbiome, and it may be associated with the status of host immune suppression. ASFV has developed a variety of mechanisms to evade host immune responses, including immunodeficiency via weakening innate immunity, blocking molecular signaling, disturbing cytokine systems and lymphoid depletion, and so on [22,32]. Although detailing these mechanisms is needed in the future, the results of the current study provide evidence for understanding the ASFV–pig immune system interaction.

Additionally, the results of this study can provide evidence for host–viral interactions and immunopathology in human VHF. Human VHF usually requires BL-3 and BL-4 facilities, and most experimental studies use rodent models [57]. On the other hand, pigs are very similar to humans in terms of anatomy and the functions of the immune system, e.g., the presence of tonsils, which are absent in rodents. The porcine immune system resembles humans for more than 80% of the analyzed parameters in contrast to mice with only about 10% [56]. For this reason, this study provides useful information to help answer questions regarding immunity in human VHF. The new evidence from this study that a gut microbiome affected by VHF infection can degrade the host’s immune function during the early stage of infection may inspire research on VHF etiology.

## 5. Conclusions

We observed dynamic changes in the gut microbiota of pigs infected with ASFV. As ASFV, which is a representative agent of animal VHF, causes severe systemic lymphoid depletion, enormous changes in symbiotic microbiota can be induced by the impaired GI immune system. Our results indicate that ASFV can cause severe perturbation of the gut microbiota, leading to a decrease in biodiversity and an increase in the relative abundance of harmful bacteria, which can affect the function of the microbiota. The predicted immune system function in the gut of ASFV-infected pigs was significantly lower than that of healthy pigs in 15 KEGG pathways. Based on these results, we provide evidence that changes in the gut microbiota during viral infection can impact disease outcomes. While the impact and mechanisms of the interaction between virus infection and gut microbiota are unclear, the microbiota may play an important role in ASF pathogenesis. Therefore, an in-depth study on the interaction between ASFV infection and the microbiome is necessary in the future.

## Figures and Tables

**Figure 1 vetsci-10-00360-f001:**
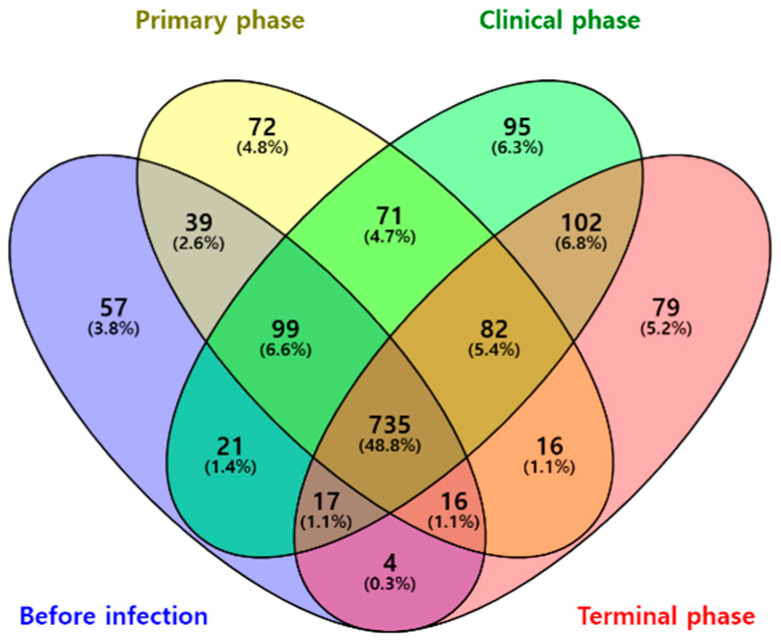
Venn diagram representing the shared and unique OTUs by different phases of ASFV infection. Numbers on the diagram represent OTUs correlated within the total sequences of each group.

**Figure 2 vetsci-10-00360-f002:**
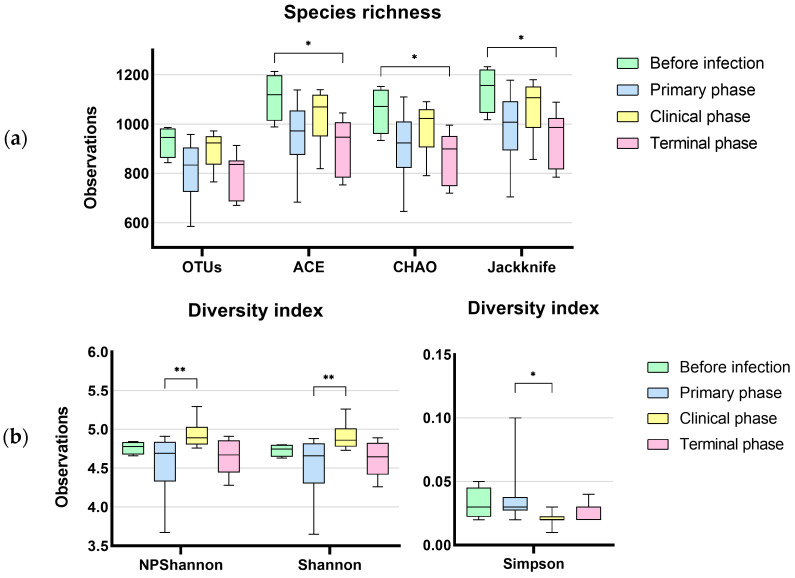
Dynamic changes in gut microbial alpha diversity of ASFV-infected pigs determined by (**a**) observed operative taxonomic units (OTUs), ACE, Chao1, and Jackknife index, (**b**) NP Shannon, Shannon, and Simpson. Statistical significance was revealed with the decrease in species richness and diversity during ASFV infection (* *p* < 0.05, ** *p* < 0.01). On the other hand, the diversity indices significantly increased at the terminal phase of ASF.

**Figure 3 vetsci-10-00360-f003:**
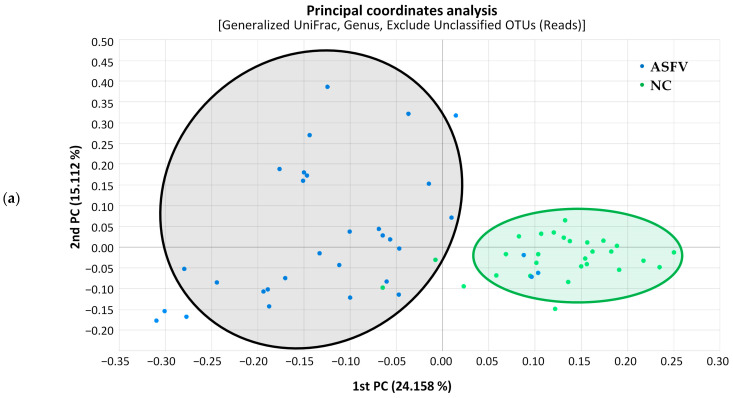
Scatterplot from principal coordinate analysis (PCoA) based on generalized UniFrac metrics between gut microbiome samples. (**a**) PCoA plot showing a distinct separation of ASFV-infected (ASFV, black) from negative control (NC, green) samples (*p* < 0.001, PERMANOVA). Principal coordinates 1 and 2 (PC1 and PC2) account for 24.158% and 15.112% of the variance, respectively, along the x and y axes. (**b**) PCoA plot revealing significant differences in bacterial composition among the four phases, which are before infection and NC (green), primary phase (blue), clinical phase (yellow), and terminal phase (red) (*p* < 0.001, PERMANOVA). Principal coordinates 1 and 2 (PC1 and PC2) account for 23.150% and 17.920% of the variance, respectively, along the x and y axes.

**Figure 4 vetsci-10-00360-f004:**
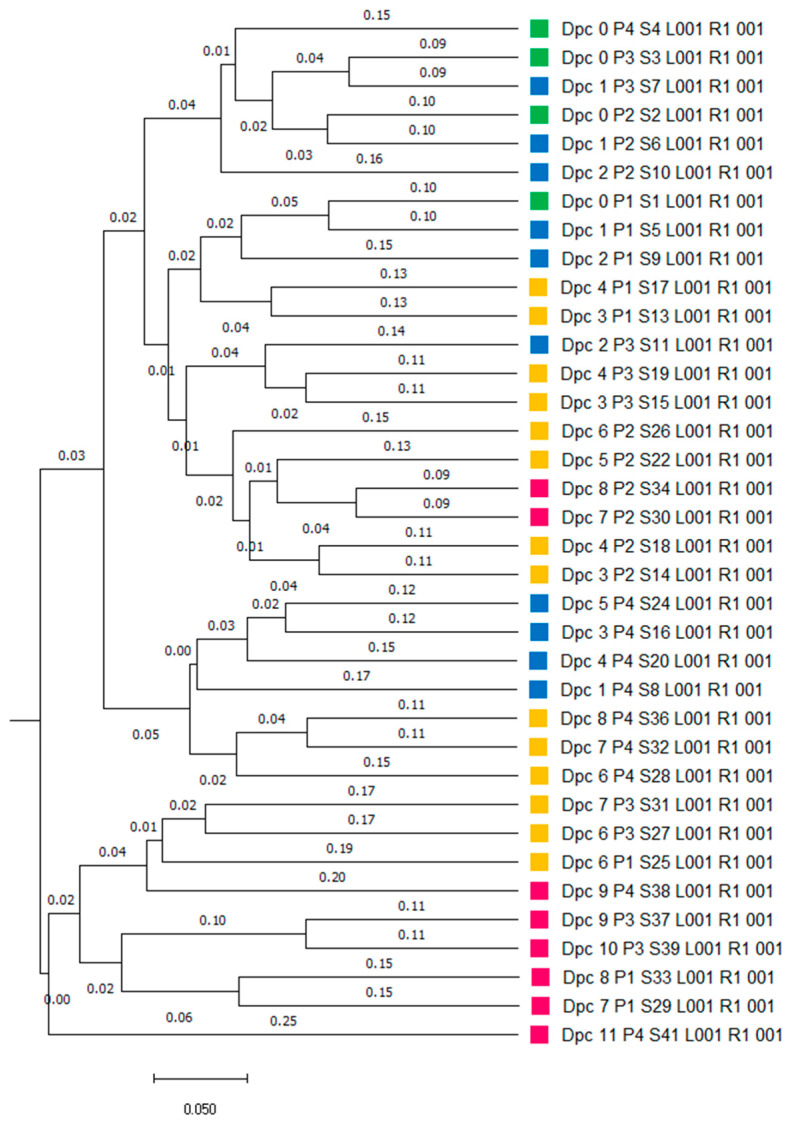
Unweighted pair group method with arithmetic mean (UPGMA) tree based on generalized UniFrac metrics among the 4 phases of ASFV infection. Box colors of the sequence represent the 4 phases, respectively (green = before infection, blue = primary phase, yellow = clinical phase, and red = terminal phase).

**Figure 5 vetsci-10-00360-f005:**
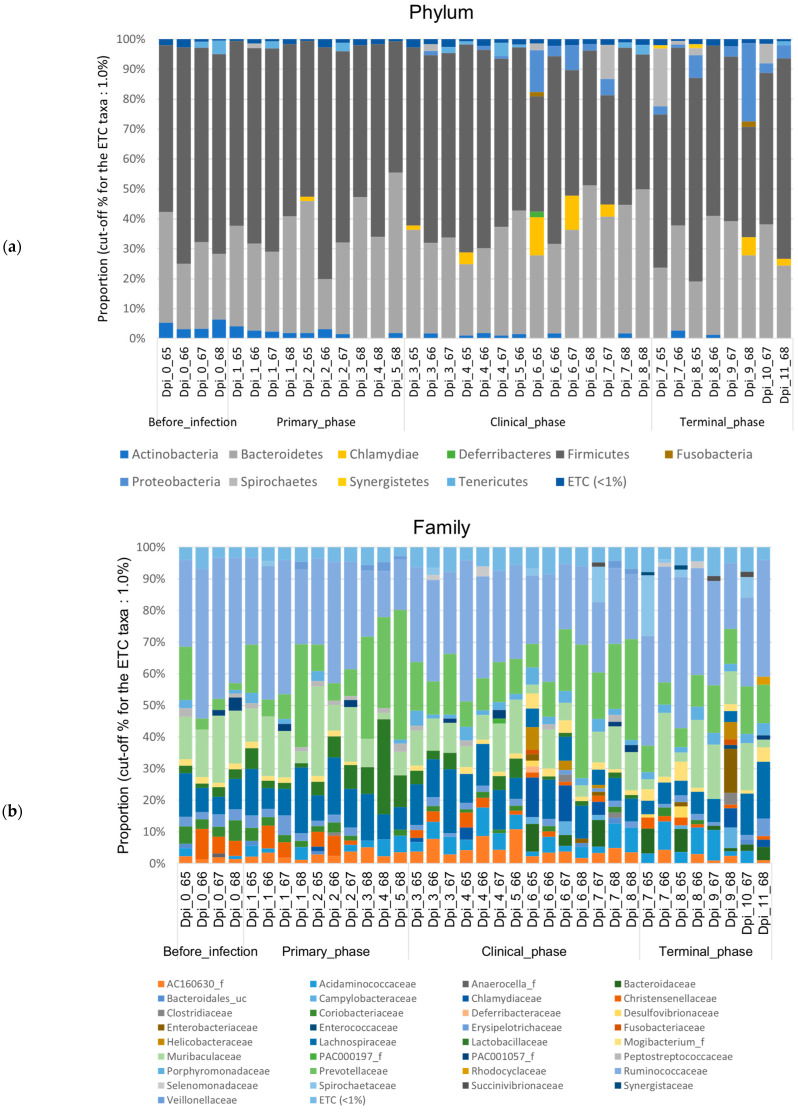
Stacked bar plots showing the relative abundance of the microbiome at phylum, family, and genus levels during the 4 phases of ASFV infection. The samples were normalized by rarefaction for compositional analysis. Only OTUs that comprise more than 1% of the total abundance are represented. (**a**) Phylum-level community bar plot, (**b**) family-level community bar plot, and (**c**) genus-level community bar plot.

**Figure 6 vetsci-10-00360-f006:**
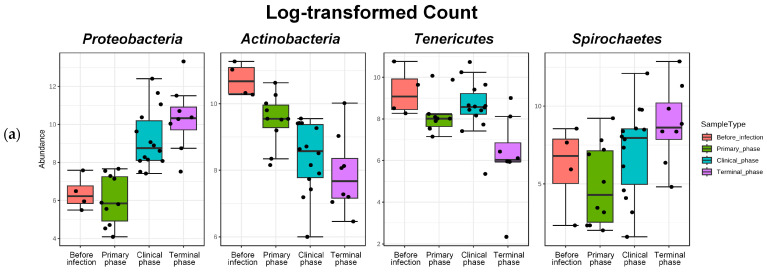
Bar plots showing the features with significant differences across the 4 phases of ASFV infection. Log-transformed count showing significant variation in their relative abundance (FDR < 0.05). (**a**) The 4 most significant phyla showing differences among the phases, as determined by DFR values, and (**b**) the top 8 families showing significant differences among the phases, as determined by FDR values. Dots, pointing to each pigs.

**Figure 7 vetsci-10-00360-f007:**
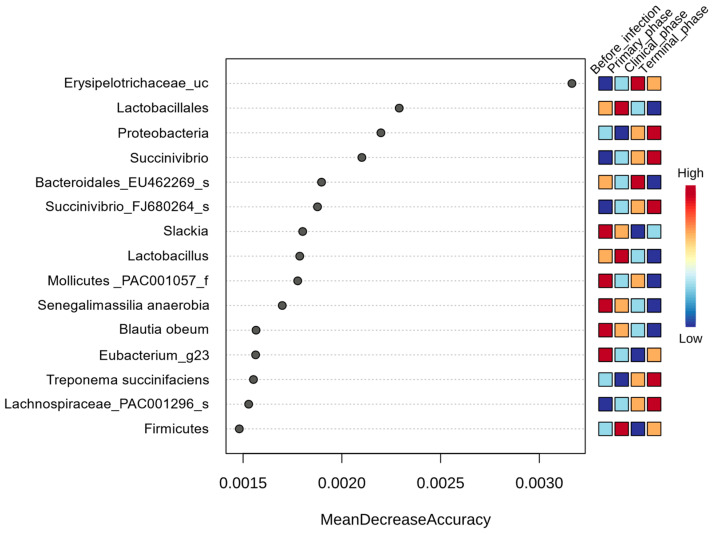
Random Forest analysis of the taxonomic differential abundance within pigs’ gut microbiota across the 4 phases of ASFV infection. For the analysis, features with at least 4 reads and 10% minimum prevalence across samples were used and the data were transformed using a centered log ratio (CLR).

**Figure 8 vetsci-10-00360-f008:**
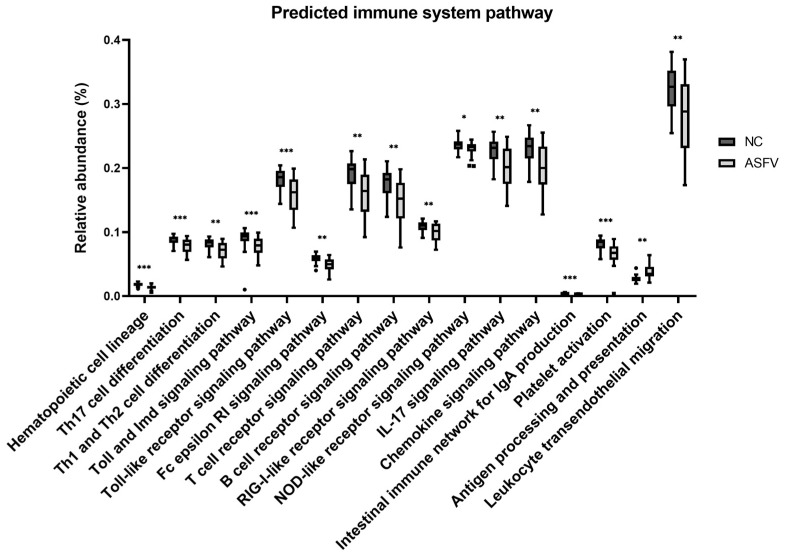
Presumptive immune functions of the gut microbiome in the negative control (NC) and ASFV-infected (ASFV) groups. Significant differences in the relative abundance of predicted metagenome profiles were found for 16 immune system pathways. (* FDR < 0.05, ** FDR < 0.01, *** FDR < 0.001).

**Table 1 vetsci-10-00360-t001:** Sample distribution by four phases of ASFV-infected pigs. During the experiment, pigs were clinically evaluated based on criteria including body temperature, appetite, recumbency, subcutaneous hemorrhage, joint swelling, and diarrhea. Fecal samples from pigs without fever or clinical symptoms during ASFV infection (incubation period) were assigned to the primary phase. Samples from pigs with fever (>40 °C) and clinical symptoms (>10% of the maximum score) were assigned to the clinical phase, and samples collected two days before death, when pigs exhibited moribundity, were assigned to the terminal phase. Outliers that had problems in the sampling process were excluded.

Phase	ASFV-Infected Pigs			
	P1	P2	P3	P4
Before infection	0 dpi	0 dpi	0 dpi	0 dpi
Primary phase	1–2 dpi	1–2 dpi	1–2 dpi	1, 3, 4, 5 dpi
Clinical phase	3, 4, 6 dpi	3, 4, 5, 6 dpi	3, 4, 6, 7 dpi	6, 7, 8 dpi
Terminal phase	7, 8 dpi	7, 8 dpi	9, 10 dpi	9, 11 dpi

**Table 2 vetsci-10-00360-t002:** Alpha diversity of gut microbiome during the 4 phases of ASFV infection. The data represent the mean ± standard deviation. Values in the same column with different superscript letters are considered significantly different (*p* < 0.05), whereas the absence of superscript letters indicates no significant differences (*p* > 0.05).

	Sampling Depth	Richness Index		Diversity Index		Good’s Coverage (%)
	Reads	ACE	Chao1	Shannon	Simpson	
Before infection	55,096	1058.47 ± 76.52 ^a^	1016.49 ± 73.01 ^a^	4.73 ± 0.06 ^a^	0.03 ± 0.01 ^a^	99.78 ± 0.01
Primary phase	51,696	917.49 ± 129.44 ^ab^	882.01 ± 124.97 ^ab^	4.54 ± 0.36 ^ab^	0.04 ± 0.02 ^ab^	99.78 ± 0.05
Clinical phase	54,088	996.82 ± 85.54 ^ab^	963.34 ± 79.98 ^ab^	4.91 ± 0.16 ^b^	0.02 ± 0.01 ^b^	99.81 ± 0.04
Terminal phase	45,985	883.79 ± 90.74 ^b^	849.29 ± 88.40 ^b^	4.62 ± 0.21 ^ab^	0.03 ± 0.01 ^ab^	99.77 ± 0.03

**Table 3 vetsci-10-00360-t003:** Presumptive immune functions of the gut microbiome in the negative control (NC) and ASFV-infected (ASFV) groups. Significant differences in the relative abundance of predicted metagenome profiles were found for 16 immune system pathways. KEGG pathways were listed at the third level, and subclasses were arbitrarily grouped based on their common characteristics.

No.	Subclass	Definition	Mean Relative Abundance (%)		*p*-Value	*p*-Value (FDR)
			NC	ASFV		
1	Cell differentiation	Hematopoietic cell lineage	0.01815	0.01390	1.74 × 10^−5^	0.000228
2		Th17 cell differentiation	0.08841	0.07897	0.000158	0.000806
3		Th1 and Th2 cell differentiation	0.08243	0.07100	0.000332	0.001345
4	Receptor signaling	Toll and Imd signaling pathway	0.09005	0.07785	2.83 × 10^−5^	0.000293
5		Toll-like receptor signaling pathway	0.18316	0.15916	0.000149	0.0008
6		FC epsilon RI signaling pathway	0.05917	0.04857	0.00023	0.001054
7		T-cell receptor signaling pathway	0.19233	0.16120	0.000555	0.001915
8		B-cell receptor signaling pathway	0.17812	0.14869	0.001539	0.003663
9		RIG-I-like receptor signaling pathway	0.10932	0.10016	0.003627	0.007403
10		NOD-like receptor signaling pathway	0.23561	0.23022	0.009057	0.014872
11	Cytokine signaling	IL-17 signaling pathway	0.22743	0.20248	0.000826	0.002402
12		Chemokine signaling pathway	0.23114	0.20199	0.001401	0.003447
13	Others	Intestinal immune network for IgA production	0.00403	0.00274	3.19 × 10^−5^	0.000301
14		Platelet activation	0.08215	0.06607	8.55 × 10^−5^	0.000585
15		Antigen processing and presentation	0.02735	0.03802	0.000332	0.001345
16		Leukocyte transendothelial migration	0.32469	0.28145	0.00431	0.008448

## Data Availability

Not applicable.

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
