# Peer review of "Alteration of the Gut Microbiota in Pigs Infected with African Swine Fever Virus"

_vetsci, 2023, doi:10.3390/vetsci10050360_

Round 1

Reviewer 1 Report

Major comments

Introduction

-        L93-96: rewrite the ‘’objective’’ of you study, more clear

Materials and Methods

-        L107-118: provide details about the experimental animals (e.g., number, sex, vaccinations)

-        L137-152: add appropriate references. 

-        L171-185: add appropriate references.

Results 

-        Figures 2 and 5 s: you should use these figures in higher analysis 

Conclusion:

-        You should expand the part of ‘’conclusion’’.

Minor comments

-        L60: The clinical course of ASF ….

-        L122:  slatted floors during 7-day

-        L133: …to induce general anaesthesia

-        L279: .. data for the NC group is referred to as supplementary materials..

-        L340: … pathways were associated with the..

Author Response

Response to reviewers’ comments

We are pleased to resubmit a revised manuscript (no. vetsci-2306711) entitled “Immunomodulatory effect of the gut microbiome altered by African swine fever virus in pigs” for reconsideration in Journal of Veterinary Sciences published by MDPI as an original manuscript. We have carefully evaluated the reviewer’s comments and have provided a point-by-point response below. Changes in the manuscript have been identified by colored font. We hope that the revised manuscript meets the reviewers’ expectations at Journal of Veterinary Sciences.

Author's Reply to the Review Report (Reviewer #1)

  • L93-96: rewrite the ‘’objective’’ of you study, more clear

Thank you for the comment. We have revised the sentence according to your comment and now clearly stated the objective of the current study.

  • L107-118: provide details about the experimental animals (e.g., number, sex, vaccinations)

Thank you for the comment. We added details about the experimental pigs, including number, sex and vaccinations.

  • L137-152: add appropriate references.

Thank you for the comment. We have reviewed your comment and found that the experimental procedure mentioned in L137-152 is not related to the current study. Therefore, we have removed the paragraph. Thank you for bringing this to our attention.

  • L171-185: add appropriate references.

Thank you for the comment. We added appropriate references according to your comments.

  • Figures 2 and 5 s: you should use these figures in higher analysis

I apologize, but I am not sure what is meant by "higher analysis." Could you please provide more details or clarification on what you are referring to? I will do my best to incorporate your feedback into our response to reviewers’ comments.

  • You should expand the part of ‘’conclusion’’.

Thank you for the comment. We expanded the part following the comment.

  • L60: The clinical course of ASF ….

Thank you for the comment. We revised the sentence following the comment.

  • L122: … slatted floors during 7-day

Thank you for the comment. We revised the sentence following the comment.

  • L133: …to induce general anaesthesia

Thank you for the comment. We revised the sentence following the comment.

  • L279: .. data for the NC group is referred to as supplementary materials..

Thank you for the comment. We revised the sentence following the comment.

  • L340: … pathways were associated with the..

Thank you for the comment. We revised the sentence following the comment.

Reviewer 2 Report

The authors studied the immunomodulatory effect of gut microbiome altered by African Swine Fever Virus (ASFV) in pigs and provided evidence for further understanding the ASFV-pig interaction. They found that changes in gut microbiome composition during ASFV infection may be associated with the status of immune-modulation.

Line 109, Please provide the experimental pigs’ information, such as age, sex, body weight and genetic background.

Line 120-124, these words should be moved to the “2.2. Animals” part.

Line 157, Please specify which days these four stages refer to in this study. Which stage do these samples belong to?

Line 205, 216, a t-test was not suitable here because these seven pigs were repeatedly measured. A GLM analysis using repeated measurements is recommended.

Figure 2: There was a problem with the statistical method, so the results may need to be modified here. The resolution of the graph is too low.

Table 1: There was a problem with the statistical method, so the results may need to be modified here.

Figure 3b: There was a problem with the annotation in the figure.

Figure 5, There was a problem with the statistical method, so the results may need to be modified. Here, the author did not consider the impact of different times on intestinal microorganisms in pigs.

Line 522-523, “Therefore, regulating the gut microbiota may be a promising method for the prevention and treatment of ASFV”. This sentence should be deleted. ASFV is not so easy to prevention and treatment.

Author Response

Response to reviewers’ comments

We are pleased to resubmit a revised manuscript (no. vetsci-2306711) entitled “Immunomodulatory effect of the gut microbiome altered by African swine fever virus in pigs” for reconsideration in Journal of Veterinary Sciences published by MDPI as an original manuscript. We have carefully evaluated the reviewer’s comments and have provided a point-by-point response below. Changes in the manuscript have been identified by colored font. We hope that the revised manuscript meets the reviewers’ expectations at Journal of Veterinary Sciences.

Author's Reply to the Review Report (Reviewer #2)

The authors studied the immunomodulatory effect of gut microbiome altered by African Swine Fever Virus (ASFV) in pigs and provided evidence for further understanding the ASFV-pig interaction. They found that changes in gut microbiome composition during ASFV infection may be associated with the status of immune-modulation.

  • Line 109, Please provide the experimental pigs’ information, such as age, sex, body weight and genetic background.

Thank you for the comment. We added the pig’s profiles, such as age, sex, body weight and genetic background in the “2.2. Animals” part.

  • Line 120-124, these words should be moved to the “2.2. Animals” part.

Thank you for the comment. We moved those words to the “2.2. Animals” part.

  • Line 157, Please specify which days these four stages refer to in this study. Which stage do these samples belong to?

Thank you for the comment. We added a table regarding the sample distribution to the manuscript.

  • Line 205, 216, a t-test was not suitable here because these seven pigs were repeatedly measured. A GLM analysis using repeated measurements is recommended.

Thank you for the comment. We understand that t-tests cannot be used for repeatedly measured samples, and we appreciate your recommendation of using a Generalized Linear Model. We have followed your suggestion and used GLM and post-hoc analysis for this purpose. Furthermore, for the differential abundance analysis, we have updated our results using statistical tests such as deseq2 and random forest, which are suitable for repeatedly measured samples. We appreciate your feedback and welcome any further comments or questions you may have.

  • Figure 2: There was a problem with the statistical method, so the results may need to be modified here. The resolution of the graph is too low.

Thank you for the comment. We have performed statistical analysis using the new methods as mentioned above, and have generated new plots that reflect the results. Additionally, we have uploaded a file with a higher resolution to the current manuscript.

  • Table 1: There was a problem with the statistical method, so the results may need to be modified here.

Thank you for the comment. We have updated the table to reflect the new statistical results.

  • Figure 3b: There was a problem with the annotation in the figure.

Thank you for your comment. We have addressed the issue and made sure that the annotation is clearly displayed.

  • Figure 5, There was a problem with the statistical method, so the results may need to be modified. Here, the author did not consider the impact of different times on intestinal microorganisms in pigs.

Thank you for your comment. We have acknowledged the issue with our statistical methods and have used the DESeq2 method to analyze changes in the relative abundance of important taxa, as mentioned earlier. We have also modified the community bar-plot to display the normalized microbial composition of individual samples. Furthermore, we have generated a new plot based on the DESeq2 analysis results at the phylum level and added it to the manuscript as Figure 6.

  • Line 522-523, “Therefore, regulating the gut microbiota may be a promising method for the prevention and treatment of ASFV”. This sentence should be deleted. ASFV is not so easy to prevention and treatment.

Thank you for the comment. The sentence has been modified in consideration of the comment you provided.

Reviewer 3 Report

Comments on vetsci-2306711

Title: Immunomodulatory Effect of the Gut Microbiome Altered by African Swine Fever Virus in Pigs

In the present study, Young-Seung Ko et al. analyzed the dynamic changes in the intestinal microbiome of the pigs infected with a highly virulent African swine fever virus (ASFV). They showed that the relative abundance of short-chain-fatty-acids-producing bacteria, such as Ruminococ caceae, Roseburia, and Blautia, was significantly decreased, while the abundance of Proteobacteria and Spirochaetes was increased during the ASFV infection. Moreover, the abundance of 15 immune-related pathways was significantly decreased in the ASFV-infected pigs as predicted by PICRUSt-based functional analysis. The work is interesting and potentially relevant. Several concerns should be addressed.

1.       Bacteriological and immunological assays should be carried out to validate the data derived from the next-generation sequencing.

2.       The presentation of the figures is poor and should be improved.

3.       The manuscript should be revised by native English speakers.

Author Response

Response to reviewers’ comments

We are pleased to resubmit a revised manuscript (no. vetsci-2306711) entitled “Immunomodulatory effect of the gut microbiome altered by African swine fever virus in pigs” for reconsideration in Journal of Veterinary Sciences published by MDPI as an original manuscript. We have carefully evaluated the reviewer’s comments and have provided a point-by-point response below. Changes in the manuscript have been identified by colored font. We hope that the revised manuscript meets the reviewers’ expectations at Journal of Veterinary Sciences.

Author's Reply to the Review Report (Reviewer #3)

In the present study, Young-Seung Ko et al. analyzed the dynamic changes in the intestinal microbiome of the pigs infected with a highly virulent African swine fever virus (ASFV). They showed that the relative abundance of short-chain-fatty-acids-producing bacteria, such as Ruminococ caceae, Roseburia, and Blautia, was significantly decreased, while the abundance of Proteobacteria and Spirochaetes was increased during the ASFV infection. Moreover, the abundance of 15 immune-related pathways was significantly decreased in the ASFV-infected pigs as predicted by PICRUSt-based functional analysis. The work is interesting and potentially relevant. Several concerns should be addressed.

  1. Bacteriological and immunological assays should be carried out to validate the data derived from the next-generation sequencing.

Thank you for the comment. Basically, microbiome analysis is conducted to see the whole gut normal flora, since most of them are difficult to culture. However, when we got results from the microbiome analysis, we cultured and confirmed meaningful bacteria, and it was not possible, we validated it through quantitative PCR.

  1. The presentation of the figures is poor and should be improved.

Thank you for the comment. As per your suggestion, we have attempted some new analysis methods and obtained new figures. In the process, we increased the resolution and also carefully considered how to appropriately represent the differences in microbiome among the comparison groups. Thank you for your feedback, and please let us know if you have any further comments or questions.

  1. The manuscript should be revised by native English speakers.

Thank you for the comment. Concerning the English editing, our manuscript has already edited in English provided by MDPI. We informed the English Editing Department of this fact via email and attached the certificate as proof. Please consider this and let us know what we should do for the manuscript.

Round 2

Reviewer 2 Report

The authors addressed all my concerns. 

Author Response

/ thank you

Reviewer 3 Report

Regrettably, the major concerns have not been addressed adequately.

Author Response

thank you